# Development of a Portland Cement-Based Material with *Agave salmiana* Leaves Bioaggregate

**DOI:** 10.3390/ma15176000

**Published:** 2022-08-30

**Authors:** Felipe Rosas-Díaz, David Gilberto García-Hernández, José M. Mendoza-Rangel, Bernardo T. Terán-Torres, Sergio Arturo Galindo-Rodríguez, Cesar A. Juárez-Alvarado

**Affiliations:** 1Facultad de Ingeniería Civil, Universidad Autónoma de Nuevo León, San Nicolás de Los Garza 66451, Nuevo León, Mexico; 2Departamento de Química Analítica, Facultad de Ciencias Biológicas, Universidad Autónoma de Nuevo León, San Nicolás de Los Garza 66451, Nuevo León, Mexico

**Keywords:** plant-based concrete, *Agave salmiana*, natural fibers, compressive strength, thermal conductivity

## Abstract

Depending on the morphology of the natural fibers, they can be used as reinforcement to improve flexural strength in cement-based composites or as aggregates to improve thermal conductivity properties. In this last aspect, hemp, coconut, flax, sunflower, and corn fibers have been used extensively, and further study is expected into different bioaggregates that allow diversifying of the raw materials. The objective of the research was to develop plant-based concretes with a matrix based on Portland cement and an aggregate of *Agave salmiana* (AS) leaves, obtained from the residues of the tequila industry that have no current purpose, as a total replacement for the calcareous aggregates commonly used in the manufacturing of mortars and whose extraction is associated with high levels of pollution, to improve their thermal properties and reduce the energy demand for air conditioning in homes. Characterization tests were carried out on the raw materials and the vegetal aggregate was processed to improve its compatibility with the cement paste through four different treatments: (a) freezing (T/C), (b) hornification (T/H), (c) sodium hydroxide (T/NaOH), and (d) solid paraffin (T/P). The effect of the treatments on the physical properties of the resulting composite was evaluated by studying the vegetal concrete under thermal conductivity, bulk density, and compressive strength tests with a volumetric ratio between the vegetal aggregate and the cement paste of 0.36 and a water/cement ratio of 0.35. The hornification treatment showed a 15.2% decrease in the water absorption capacity of the aggregate, resulting in a composite with a thermal conductivity of 0.49 W/mK and a compressive strength of 8.66 MPa, which allows its utilization as a construction material to produce prefabricated blocks.

## 1. Introduction

Since the second half of the 20th century, global economic dynamics have led to important social developments related to negative environmental indicators that stand out in parallel due to anthropogenic activities [1,2]. Globally, it is estimated that the construction industry is responsible for 30–40% of energy consumption directly during building construction, usage, and demolitions, and indirectly through the production of materials used (embodied energy) [3], and 40–50% of global greenhouse gas emissions [2,3,4,5]. In addition, building construction annually consumes 40% of stone (such as sand and gravel), 25% of virgin wood, and 16% of water and it is responsible for the generation of 30% of the world’s solid waste [2,6].

The phases of a building project can be divided according to the processes during its life cycle (LCA) which can be seen in Figure 1 [3].

Several studies have concluded that the use phase in the LCA of buildings contributes to the largest environmental impact, due to its extended duration and related to the use of fossil fuels for the supply of electricity and heating [3]. As shown in Figure 2, for three energy consumption scenarios, this phase consumes the largest amount of energy up to values exceeding 80% [7], however, as the annual energy consumption decreases, the energy contained in the materials starts to become more relevant to the project [3].

Therefore, the construction sector is facing important challenges that bring about new research for the development of materials with a lower environmental impact by essentially using vegetal aggregates to manufacture environmentally friendly composites, achieving the preservation of raw materials, the reuse of residues, and energy cost savings [8,9,10].

Materials made from plant-based raw materials are the most promising for construction due to their potential applications in composite materials [11]. One example is their use as a replacement for concrete components, due to the advantages they can bring to these composites as shown in Table 1, making it a more sustainable construction material, and leading to the development of plant-based concrete [12,13].

A plant-based concrete consists of a mixture of lignocellulosic granules and a mineral binder, which leads to improved effects on the hygrothermal properties (lower thermal conductivity) and lower environmental impacts. The mechanical capacity of plant-based concrete depends on the binder and the amount of fiber used, its geometry and distribution, and adhesion to the matrix [9]. Although the strength is lesser than the usual construction material, the best results in its development are shown in the lowest possible values for thermal conductivity, and the higher compressive strength values, with a minimum value of 0.2 MPa [14].

Various sources of bioaggregates have been used, such as wood, coconut, sisal, palm, bamboo, and bagasse, among others. However, it is necessary to diversify the natural fibers used by considering local raw materials in each region (e.g., *Agave* in Mexico), to contribute to the energy efficiency of buildings by incorporating sustainability aspects in the materials. It is relevant to increase research in the area by incorporating a greater diversity of plant and cementitious materials to expand knowledge, obtain better results and, at the same time, create a promising area of application for the development of research projects aimed at identifying other fields in which they can be used [15].

To contribute to the solution of the problem, this study evaluates the incorporation of lignocellulosic aggregates from the *Agave salmiana* leaves, obtained as a residue from the distillates industry in Mexico, as a total replacement of fine limestone aggregate in the production of plant-based concrete, with mechanical properties that meet the requirements for non-structural applications and with improved thermal insulation properties. The lignocellulosic aggregates received four treatments to evaluate their effect on the physical and mechanical properties of Portland cement-based concretes in thermal conductivity tests, calculation of bulk density, and mechanical compression strength.

## 2. Materials and Methods

The procedures implemented for the development of the research are based on the current regulations to produce mortars, which are described below.

### 2.1. Materials

#### 2.1.1. Cementitious Material

The cementitious material corresponds to Portland cement type III according to ASTM C150 classification, and has a density of 3.04 g/cm^3^ [16], with an average particle size of 24.58 µm [17,18]. The chemical composition was obtained by mass-energy dispersive XRF analysis under a Helium atmosphere in a PANalytical-Epsilon 3 and the results are shown in Table 2.

#### 2.1.2. Plant Aggregate from *Agave salmiana* (AS) 

Morphological characterization: The morphology of the AS used to produce distillates is shown in Figure 3, and its utilization is mainly focused on agave head distillation (i.e., leave base). This study worked with AS leaves, which are considered waste and therefore do not have well-defined disposal (Figure 3, right). To determine whether it was necessary to select some of the components of the leaf in the fabrication of the mortar, it was separated into three parts: pith (leaf medulla) [M], cuticle (leaf epidermis) [C], and pith with cuticle [MC].Chemical characterization: The chemical analysis of the fiber was performed according to TAPPI (Technical Association of the Pulp and Paper Industry) procedures and experimental methods, described by Wise L.E [19], and Rowell [20], to determine the ash content [21], ethanol-toluene extractable [22], acid-insoluble lignin content [23], holocellulose content [19], α-cellulose, and hemicellulose [20,24,25,26,27].Physical characterization:

Sample preparation: The procedure is shown in Figure 4. At this stage, the AS was manually cut (Figure 4a), then the pith (M), cuticle (C), and pith with cuticle (MC) were separated (Figure 4b) and dried in an oven with the air circulation at a temperature of 60 ± 5 °C until reaching a constant mass (variation of less than 0.1% between two readings at 24 h) (Figure 4c) [28]. Subsequently, the dried sample was sieved, leaving only the fraction retained between sieve No. 4 (4.75 mm) and No. 30 (0.6 mm) and stored in airtight bags (Figure 4d). After drying, the initial water content was determined in relation to the dry weight.Water absorption of the vegetable aggregate: 25 g of each AS sample (M, C, and MC) were weighed and immersed in distilled water for 1, 15, 240, 1440, and 2880 min. Subsequently, they were extracted and centrifuged at 120 RPM for 50 s [28,29]. The water absorption was calculated through Equation (1) [8]:
(1)$$Absorption\;\left(\%\right)=\frac{W_{W}-W_{D}}{W_{D}}\times 100$$
where *W_W_* is the weight after water immersion and surface drying by centrifuge and *W_D_* is the initial weight after oven drying.

3.Bulk density of the vegetable aggregate: This was determined by placing a sample of each type (M, C, and MC) in a cylindrical container of known dimensions, which was shaken 10 times to subsequently determine the volume occupied by the fiber and its mass, thus obtaining the apparent density of the vegetable aggregate [28,30].4.Thermal conductivity of the fiber: Thiswas performed using a Tempos, Meter Group brand, based on the hotwire technique that complies with the specifications of the IEEE 442-1981 standard and ASTM D5334-08. The data were collected in a 10 min test time using the KS-3 probe [28].

**Figure 3 materials-15-06000-f003:**
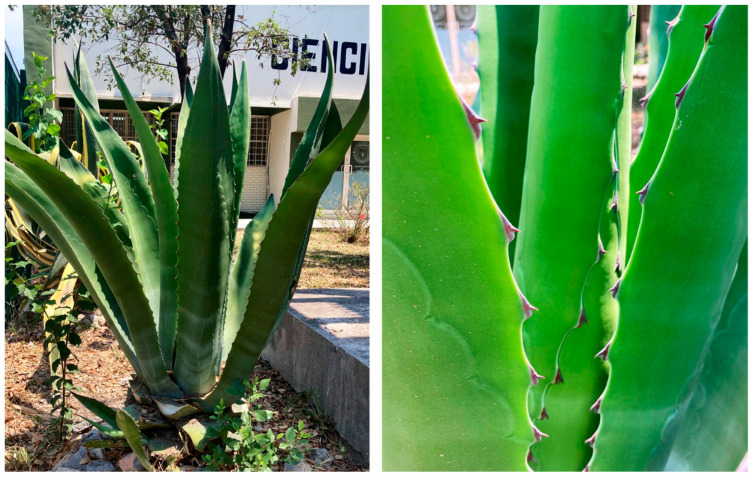
(**Left**) *Agave salmiana* plant. (**Right**) Detail photograph of the *Agave salmiana* leaf.

**Figure 4 materials-15-06000-f004:**
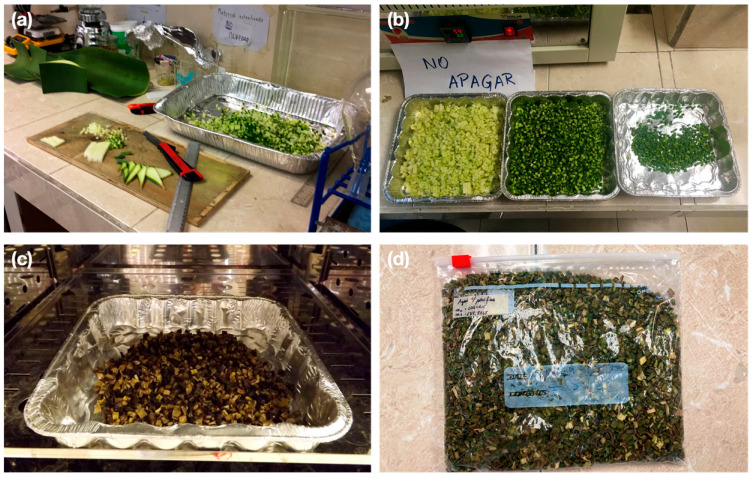
Bioaggregate preparation procedure. (**a**) Manual fiber cutting; (**b**) Separation of parts; (**c**) Oven drying; (**d**) Fiber storage.

### 2.2. Treatments to AS Aggregate

To improve the physical properties of the composite, the aggregate used was subjected to different treatments reported in the literature to improve the compatibility between the matrix and the bioaggregate. These treatments are described below and can be seen in Figure 5.

#### 2.2.1. Untreated Plant Aggregate [S/T]

An untreated control sample was used. For this, a sample of the oven-dried plant aggregate was taken at 60 ± 5 °C and subsequently stored in hermetic polypropylene bags. The samples were composed without distinguishing between pith, cuticle, or pith with cuticle.

#### 2.2.2. Freezing Treatment [T/C]

The natural fibers were placed inside vacuum plastic bags and then subjected to a temperature of −20 ± 5 °C. Subsequently, prior to their use, the fibers were removed from freezing and kept until they were at laboratory ambient temperature, i.e., 25 °C. They were then oven-dried at 60 ± 5 °C until a constant mass was obtained.

#### 2.2.3. Physical Treatment of Hornification [T/H]

The aggregate was placed in a container of water, where it remained for 3 h. Then, it was introduced into an oven with airflow, where it remained for 16 h at a temperature of 80 ± 5 °C. Afterward, the sample was conditioned at 23 ± 5 °C to avoid a possible thermal shock. This whole procedure corresponds to one cycle, which for the treatment was repeated ten times according to the established protocol [31,32,33].

#### 2.2.4. Chemical Treatment of NaOH [T/NaOH]

The plant aggregate was introduced into a 1% solution of sodium hydroxide that was already boiling at 95 °C and kept for one hour in a space conditioned with a gas and vapor extractor. Subsequently, the fiber was deposited in a desiccator for 30 min to gradually attain the ambient laboratory temperature. Then, the aggregate was washed with distilled water, centrifuged (100 revolutions in 50 s), and dried in the oven at 60 ± 5 °C until a constant mass was obtained.

#### 2.2.5. Solid Paraffin-Based Coating Treatment [T/P]

The solid paraffin was heated homogeneously to 100 °C, then the dry fiber was immersed for 5 min. Afterward, it was placed inside an oven at 105 °C for 15 min with absorbent paper to remove the excess of paraffin, reduce the formation of paraffin lumps in the fibers, and improve disintegration before blending [29,34].

#### 2.2.6. Physical Characterization of Treated Aggregates

After the application of the different treatments, physical characterization tests were carried out to determine water absorption with submersion times of 15, 240, 1440, and 2880 min [8,14,28,35], bulk density [28,36], and thermal conductivity of each aggregate by the transient line heat method with Meter environment equipment using a KS-3 sensor with heating and cooling cycles of 10 min [28,37].

### 2.3. Plant-Based Concrete

#### 2.3.1. Specimen Fabrication

The procedure for the manufacture of mixtures was carried out in accordance with the study conducted by Véronique Cerezo [38]. By placing the aggregate in the mixer vessel and homogenizing for one minute, the pre-mix water was added for one minute, keeping the mixer always running. Homogenization continued for one more minute and then the cementitious material was added for one minute and stirred for two minutes before incorporating the reaction water for one minute. Subsequently, stirring continued at medium speed for 4 more minutes. The total process lasted eleven minutes. The dosage is shown in Table 3 considering a volumetric ratio between the plant aggregate and the cement paste of 0.36 [38].

The 50 mm cubic specimens were considered, and they were fabricated in two layers, each tamped 32 times to ensure a smooth surface finish. Curing consisted of leaving the specimens in ambient conditions of 23 ± 2 °C and 50 ± 5% relative humidity for 7 and 28 days [39]. For the corresponding analyses, the plant-based concrete (CV) samples were immersed in acetone for 48 h to stop the hydration process of the cement paste. They were then dried under vacuum in an oven at 50 °C for 24 h [40,41,42,43].

#### 2.3.2. Bulk Density

It was determined by dividing the mass of the cubic specimens in grams by the average of the volume obtained from two measurements of the width, height, and length of the cubic specimens as stipulated in the ASTM C138 standard [44].

#### 2.3.3. Thermal Conductivity

For the thermal conductivity tests, three 40 × 40 × 160 mm specimens were manufactured with the same dosages as described in the previous section, they were cut in half and cured for 28 days. The measurements were determined using the transient line heat source method with a Tempos thermal properties analyzer manufactured by the Meter Environment brand. The heating and cooling cycle was defined in 10 min, achieving an accuracy of ±0.01 W/mK according to the nature of the samples and the manufacturer’s specifications. The RK-3 sensor was used, which was introduced with thermal paste to eliminate the air that may be between the probe and the sample to avoid altering the results [11,37].

#### 2.3.4. Compressive Strength

Compressive strength tests were performed on samples at ages 7 and 28 days. The compression test was performed under ambient conditions of 20 ± 2 °C and 50 ± 5% relative humidity with a displacement-controlled loading rate of 5 mm per minute using an INSTRON 600DX hydraulic testing machine [14,45].

#### 2.3.5. Thermogravimetric Analysis of Cement Paste 

The fragments resulting from the compression test were immersed in acetone for 48 h to stop the concrete hydration process. They were then dried under vacuum in an oven at 50 °C for 24 h [40,41]. The samples were analyzed by thermogravimetry assay using TA Instrument Calorimetry (SDT Q600, USA) in a temperature range from 25 °C to 900 °C at a heating rate of 10 °C/minute in an N_2_ atmosphere.

#### 2.3.6. Morphological Analysis

Microscopy of the plant-based concrete was performed for each of the samples under study with the different treatments [29], mounting the samples in a Petri dish to be observed under an inverted light microscope at 4× magnification. The vegetable concrete sample with the hornification treatment was analyzed by SEM scanning electron microscopy on resin-embedded (polychrome varnish) and polished specimens. Images were taken at an accelerating voltage of 15 kV, with magnifications from 20× to 500×.

## 3. Results

### 3.1. Raw material Characterization

#### 3.1.1. Morphological Characterization

The AS leaves were obtained directly and analyzed as shown in Figure 6, distinguishing three parts of the leaf: Cuticle, Pith, and Pith with Cuticle.

#### 3.1.2. Chemical Characterization of Plant Aggregate

The results obtained in the A. salmiana sample are shown in Table 4. The results of chemical characterization showed that the natural fiber is mainly constituted by cellulose with an average value of 49.64 ± 0.83% like the value of hemp or sunflower [30]; in addition, holocellulose was found in the range of 42.64 ± 1.10% and lignin at 10.61 ± 0.24% content.

An important aspect is the high content of polar compounds that were extracted with the ethanol–toluene solution, equivalent to 23.47%, compared to other fibers such as eucalyptus husk with 4.19%, which is equivalent to 23.47% [37], or oat straw with 8.70%, whose values have been reported in the literature [46]. It is important to consider this background because water has the potential to extract water-soluble compounds that could interfere in matrices by interfering in the hydraulic reaction of cementitious materials.

#### 3.1.3. Physical Properties

The results can be seen in the graph in Figure 7. The results show that the pith absorbs 24.13% more water than the cuticle. This agrees with what is described in the scientific literature that attributes the high levels of water absorption and retention to the porosity and internal structure of the plant fibers [30].

Although this factor is adverse because of the interaction it may have with the reaction water and/or with the mix in the fabrication of concrete, in conjunction with the extensive drying times required [29]. At the same time, it should be taken into consideration that porosity is the main characteristic that provides the insulating properties to the natural fiber, which makes it attractive for its use as a construction material. Nevertheless, the cuticle possesses a similar insulating performance. The bulk density (ρ) for each of the constituent parts of the agave leaf is described in Table 5.

The cuticle has a bulk density 79.39% higher than the pith. As this parameter is related to the porosity of the fiber due to the air content it may contain, these results are related to the values of thermal conductivity (λ), described in Table 5, and water absorption capacity. The results show that the cuticle has a thermal conductivity 18.77% higher than the pith and this is related to the difference in density between both components. This conductivity value is in the range reported in the literature for fibers such as hemp (0.048–0.12 W/mK), flax (0.038–0.075 W/mK) [8,12,14,47], or straw bales (0.067 W/mK) [48], which correspond to natural fibers that have been used in the development of materials with insulating properties, therefore, *Agave salmiana* has potential properties for its use in the development of construction materials.

### 3.2. Characterization of the Vegetal Aggregate after Treatments

#### 3.2.1. Water Absorption

The results of the water absorption test for the plant aggregate after being subjected to the different treatments under study are shown in Figure 8. The exposure of the aggregates to treatments allows modifying of fiber parameters that favor a greater hydrophobicity with respect to the reference sample (S/T). According to the results of the graph, it can be observed that the water absorption of the plant aggregate subjected to the treatment under study decreased between 15.03 and 20.22%, like the values obtained in treatments applied to other fibers [34]. In general, it is observed that in the case of the fiber subjected to hornification, sodium hydroxide, and paraffin treatment, the resulting curve shifts downward in the graph, so that the behavior occurs similarly to that of the untreated fiber, but reduces water penetration.

In the case of the freezing treatment, as mentioned above, the particles acquire morphological stability due to the breakdown of structural components, therefore, they retain air spaces in their interior where, in the first 500 min of immersion in water, the velocity is higher than that of the other treatments [14].

The reduction of the absorbent capacity of the raw material will have a direct effect at the time of manufacturing the plant-based concrete because it will interfere less with the generation of hydration products in the cementitious matrix and thus leading to better mechanical behavior of the compressive strength of the resulting composite. This is because, at the time of manufacturing the material, when the anhydrous components and water are mixed, the plant particles absorb and retain a large amount of water that will no longer be available for the hydration of the binder. To avoid this problem and to obtain suitable workability, the amount of water added is greatly increased, but this leads to a larger drying time after demolding [29].

Another aspect in which the reduction of the water absorption capacity could be beneficial is the durability of the fiber since the literature indicates that the interaction of the lignocellulosic material with alkaline solutions allows its degradation over time and therefore, depending on the format in which the fiber is used and the properties to be obtained in the developed composite, the reduction of this parameter would allow better performance over time despite the exposure of the lignocellulosic material to alkaline solutions [49].

#### 3.2.2. Thermal Conductivity and Bulk Density

Figure 9 shows the results of thermal conductivity and bulk density, as well as the directly proportional relationship between both parameters [8]. The existing correlation is because of the retention capacity of small airtight cavities that dissipate heat propagation by conduction due to the low thermal conductivity of air of the order of 0.028 W/mK [50].

We can divide the thermal conductivity results into those that increase the bulk density and those that decrease it. In the first group, in the case of the hornification treatment and the freezing treatment, it is observed that the vegetal aggregate’s bulk density was reduced by 15.05%, which is consistent with the greater uniformity of sizes presented in the fiber after the treatment. This behavior can be explained by the rigidity of the polymeric structure of the fiber cells produced during the wetting and drying process. In this case, the cellulose chains of the polysaccharides are arranged in a more compact form with the elimination of water during the drying process and, therefore, the microfibrils bind together in the dry state as a result of the greater packing but retain air in their interior which allows this phenomenon to occur [31]. The results of thermal conductivity on the plant aggregate after the application of treatments show that, in the case of the treatment with sodium hydroxide, the thermal conductivity of the aggregate increased due to the displacement of the air contained in the particles by the reduction of their size. Something similar occurred for the paraffin treatment, where the air was displaced by a material with thermal conductivity in the order of 0.25 W/mK, which is conducive to increasing its overall value. In the case of the freezing and hornification treatment, its high porosity in relation to the reference sample (i.e., without any treatment) is responsible for the decrease in the thermal conductivity values reported in the graph.

### 3.3. Characterization of Vegetal Concrete

#### 3.3.1. Bulk Density

The results obtained for the plant-based concrete (PBC) specimens fabricated with aggregates subjected to the different treatments under study are shown in Figure 10.

The bulk density of the samples is related to the bulk density of the plant aggregate and the densification of the cementitious matrix. An important fact that results from the fabrication of PBC using AS aggregate is the significant amount of saponins and water-soluble compounds that can extract the water when in contact with the aggregate and it can be clearly observed at the time of production of the mixture, and it is also observed in the saponin foam formed by the process shown in Figure 11a.

Figure 11a shows the effect of the saponin present in the plant aggregate, which is extracted by the reaction water of the cement and generates air bubbles that are retained in the cementitious matrix, reducing the densification of the mortar and its mechanical properties (compressive strength). In the same figure, a sample of plant-based concrete with S/T aggregate shows the resulting porosity due to the presence of air bubbles. However, in Figure 11b,c, a specimen of plant-based concrete with T/H aggregate is observed, where the paste has achieved adequate densification. This is because during the manufacturing process air bubbles were not presented as in the other case studies. If the bulk density of the plant-based concrete is compared with standard mortars or concretes (i.e., ρ = 2.5 g/cm^3^), it can be observed that with the incorporation of the plant aggregate, this parameter decreases considerably due to the apparent density of the aggregate particles [51].

#### 3.3.2. Thermal Conductivity

The thermal conductivity results can be seen in Figure 12. In general, the incorporation of lignocellulosic aggregate in the manufactured concrete has reduced the bulk density of the material to a considerable extent and, consequently, the thermal conductivity has been improved [51]. According to the results shown in Figure 11, it can be observed that the thermal conductivity of the specimens is related to the bulk density obtained. This phenomenon has been reported by several authors who have used different plant aggregates [4,14].

It can be observed that the hornification treatment produced the highest thermal conductivity values in the order of 0.49 W/mK, which corresponds to 277.80% more than the reference sample (i.e., without treatment). It is also observed that the general behavior of the samples with the different treatments can be attributed to the lower amount of bubbles as a result of the action of saponins due to the individual effect of each treatment. However, the thermal conductivity levels achieved are significantly improved in comparison with conventional mortars, for which the literature stipulates a thermal conductivity of 1.4 W/mK [50]. The relationship between the bulk density of the specimens and the thermal conductivity is shown in Figure 13.

Figure 13 shows that the physical properties of cementitious-based matrices reinforced with natural plant fibers are highly influenced by the fiber content. It can be observed that the thermal conductivity of the composites increases with higher density, this shows a direct relationship between composite density and thermal conductivity. The increase in air voids leads to a decrease in the density of the resulting materials, this results in higher thermal resistance and lower thermal conductivity [11,29,45].

#### 3.3.3. Compressive Strength

The results are shown in Figure 14. Although the density and bulk density results show that the fabricated specimens with the hornification treatment have physical properties with higher values due to the lower amount of voids created by the incorporation of the aggregate with this treatment [29,52], the mechanical properties showed a variation concerning the reference case (i.e., aggregate S/T), which has a compressive strength of 0.22 MPa, reaching 8.68 MPa with this treatment.

#### 3.3.4. Thermo Gravimetric Analysis of the Cement Paste

Figure 15 comparatively shows the results for the thermo-gravimetric analysis (TGA) of cement pastes of plant-based concretes produced with AS aggregate with different treatments. The TGA of the concrete specimens provides a quantitative evaluation of the weight loss due to endothermic reactions such as C-S-H dehydration, CH decomposition, and CaCO_3_ decomposition. The curves in Figure 15 show that for the endothermic reaction taking place at 125–400 °C due to dehydration of C-S-H and minority aluminate and sulfoaluminate phases (AFm and AFt), the weight loss was higher in the plant concrete specimens with the aggregate with the T/H treatment with a 5.22% decrease in mass, which indicates the presence of more C-S-H gel in these specimens.

This is evidenced by the results obtained on the compressive strength, where the T/H samples stood out, and in the case of the specimens with T/P treatment, although the results are significantly lower than those obtained for the sample with T/H treatment, the results obtained are 89.34% higher than those of the S/T reference samples [53]. For the endothermic reaction that occurs between 420–600 °C and that is attributable to the decomposition of Ca(OH)_2_ where water is released and CaO remains, we can observe that the sample with T/H obtained the highest mass variation with a reduction of 3.19%, which differentiates it from the other samples with respect to the hydration products that were formed and that provide it with better compressive strength properties. The same can be observed concerning the loss of mass because of the decomposition of CaCO_3_, where CO_2_ is released in the form of gas at temperatures between 600–730 °C, which indicates a larger presence of this compound with a mass reduction of 4.32% [53].

#### 3.3.5. Morphological Observation

Figure 16 and Figure 17 shows the interfacial zone between the vegetable aggregate with different treatments and the Portland cement-based matrix.

According to the images obtained from the microscopy performed on the plant-based concrete samples, the untreated sample shows a uniformly distributed porosity throughout the matrix due to the action of the saponins, in addition to a notorious detachment of the fiber that generates an air space. The interfacial zone between the OPC matrix and the bioaggregate particles is characterized by a high porosity, very frequently forming a space (detachment) around the fiber as observed in Figure 17b. The thickness of the transition zone increases with the water-cement ratio and can be explained by the higher drying shrinkage of the plant fibers.

According to the literature, a greater amount of portlandite is deposited in the transition zone and the size of this cavity is directly related to the water/cement ratio used, decreasing as the curing time increases and the smaller the difference in desiccation shrinkage of the plant fibers with respect to the cement paste [49]. The cementitious material hydrolyzes and mineralizes the plant fibers. This hydrolysis mainly causes the loss of hemicelluloses, but does not significantly affect lignin, this is due to the gradual precipitation of calcium hydroxide particles on the surface and/or in the lumen of the fibers [25].

Figure 16b shows a plant-based concrete specimen with hornification treatment where it is observed that due to the elimination of hydrosolubles, as well as the morphological stability generated by the treatment, a significant adhesion is obtained in the area between the particles and the matrix, which is in agreement with the results of the physical characterization. The same is shown in detail in Figure 17a, where if compared with the control sample (S/T) in Figure 17b, it is observed that the detachment between the matrix and the aggregate is lower, which is detrimental to the reduction of the thermal conductivity of the resulting composite, but at the same time allows obtaining better results in compressive strength.

Furthermore, in Figure 16, samples (c) and (d) corresponding to plant-based concretes fabricated with aggregate subjected to NaOH and paraffin treatment, respectively, showed good adhesion of the aggregate matrix due to the surface modification of the particles and the volumetric stability obtained. However, because they retained water-soluble compounds, constant air bubbles were observed in the matrix, which had an impact on the results.

The decrease in the absorption capacity of the fibers allows the precipitation of organic compounds to occur only on the surface of the aggregate particles. Subsequently, as the curing of the samples increases, a “pumping” effect occurs due to the increase and reduction of the particle size, allowing the densification of the surface and the lumen of the fibers with high-alkalinity products [25].

The results obtained show that the hornification treatment allows a more effective hydration of the cement due to the levels of compressive strength achieved. This can be explained by the fact that the hornification treatment stipulates 10 wetting and drying cycles, which allows a progressive washing process of the aggregate, this allows achieving the extraction of water-soluble compounds, in addition to the consequences on the morphological stability of the particles when exposed to the water involved in the manufacturing process of the plant-based concrete. These combined factors allow the *Agave salmiana* aggregate to behave in a more inert manner in the mixing process with the other components for the fabrication of the plant-based concrete and therefore allow favorable conditions for the adequate hydration of the cement, which leads to optimal results of mechanical resistance. The high value obtained allows establishing this treatment as highly effective with respect to the other treatments under study since, as mentioned above, this type of mortar is not focused on structural applications, so the minimum acceptable strength is 0.2 MPa. By having approximately 30% of the volume occupied by the aggregate and strength extensively higher than the rest of the cases, it is considered that this treatment deserves to be studied because it allows the incorporation of a larger volume of plant aggregate, further decreasing the amount of cement paste required and improving the values of bulk density and thermal conductivity.

## 4. Conclusions

The thermal conductivity and water absorption values for the plant aggregate show that, although the cuticle has a higher conductivity, it has a lower water absorption capacity, and thus the best option for its utilization in the development of composites is the non-exclusion of any of these components of the leaf. In this manner, the raw material can be better utilized by incorporating it completely into the mineral matrix.

The evaluation of the treatments facilitated the establishment of the properties of the vegetable aggregate, where important morphological changes were observed with respect to the reference sample without treatment, along with a significant decrease in the water absorption capacity, which is beneficial to promoting adequate hydration of the Portland cement. The results of the evaluation of the vegetable aggregate incorporated in the cementitious matrix are categorical in pointing out that the hornification treatment is the most adequate for the manufacturing of vegetable mortars using *Agave salmiana* leaves particles, which is directly attributable to the washing that occurs with the repetition of wetting and drying cycles hydrolyzing and extracting water-soluble compounds that interfere in the reaction of the Portland cement and that significantly reduce the resistance of the compounds using the other treatments under study.

The incorporation of a volumetric ratio between the plant aggregate with T/H and the cement paste of 0.36, allows a reduction in the density and thermal conductivity of the material, reaching values of 1.55 g/cm^3^ and 0.49 W/mK, which improves its thermal performance, resulting in a higher level of insulation for housing construction usage.

The mechanical resistance values of the developed composite are 8.68 MPa, which is well below the strength a structural member would have, therefore, an application for the composite as a surface material for envelopes must be considered. The high availability and low final valuation of Agave leaves from the Tequila industry may constitute a good opportunity to produce bioaggregates for the manufacturing of sustainable mortars for thermal insulation in construction.

## Figures and Tables

**Figure 1 materials-15-06000-f001:**
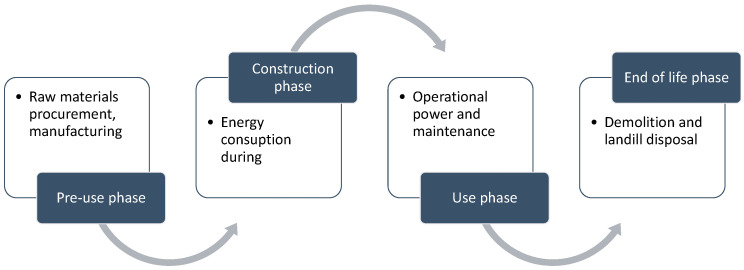
Limits of cradle-to-grave system phases in building LCA research.

**Figure 2 materials-15-06000-f002:**
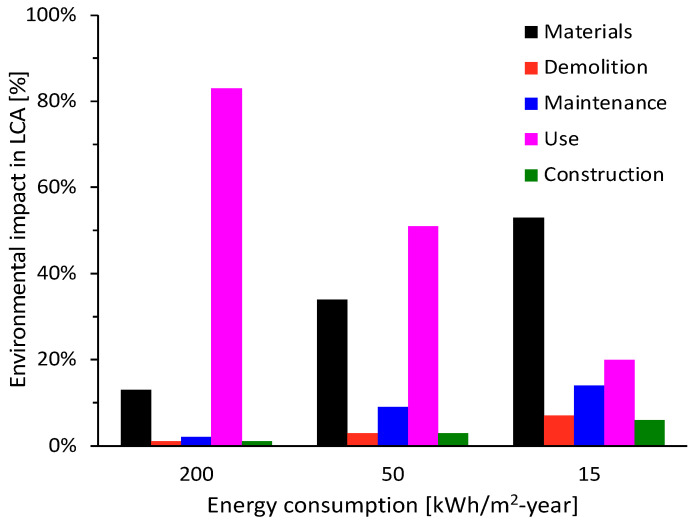
Environmental impact for the different phases of a project, depending on its energy performance.

**Figure 5 materials-15-06000-f005:**
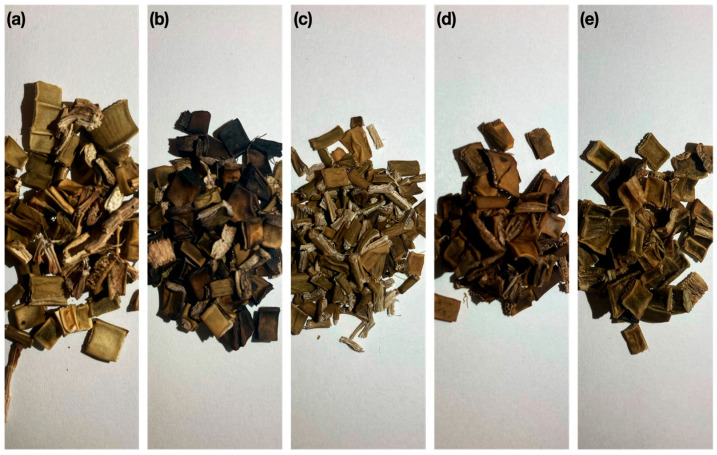
Visual appearance of the aggregate with each of the treatments. (**a**) S/T; (**b**) T/C; (**c**) T/H; (**d**) T/NaOH; (**e**) T/P.

**Figure 6 materials-15-06000-f006:**
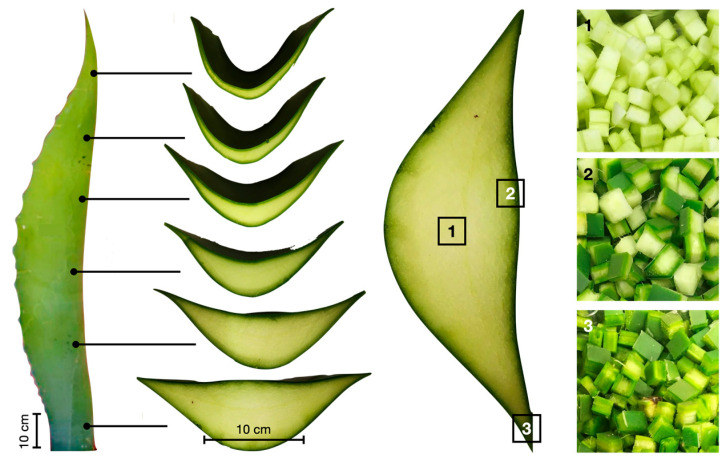
Morphology of an AS leaf and cross-section of the leaf showing (**1**) pith, (**2**) cuticle with pith, and (**3**) cuticle area.

**Figure 7 materials-15-06000-f007:**
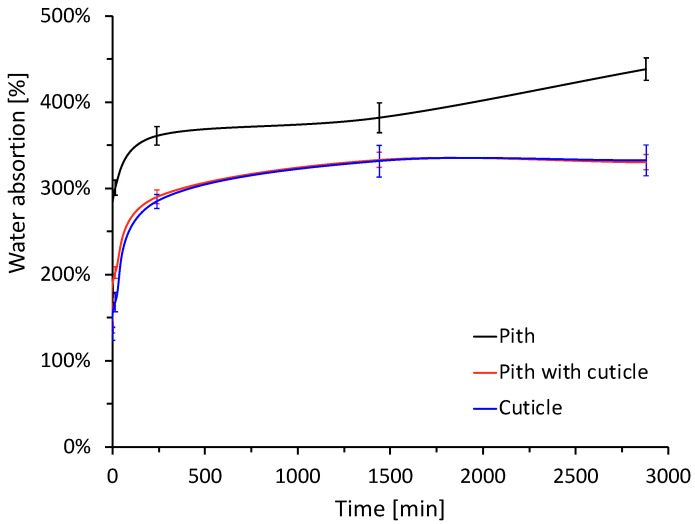
Water absorption of the plant aggregate is under study.

**Figure 8 materials-15-06000-f008:**
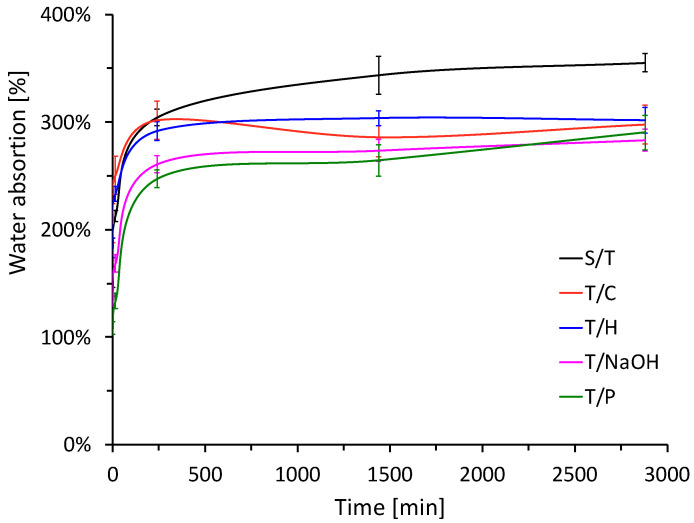
Water absorption for the plant aggregate subjected to the different treatments under study.

**Figure 9 materials-15-06000-f009:**
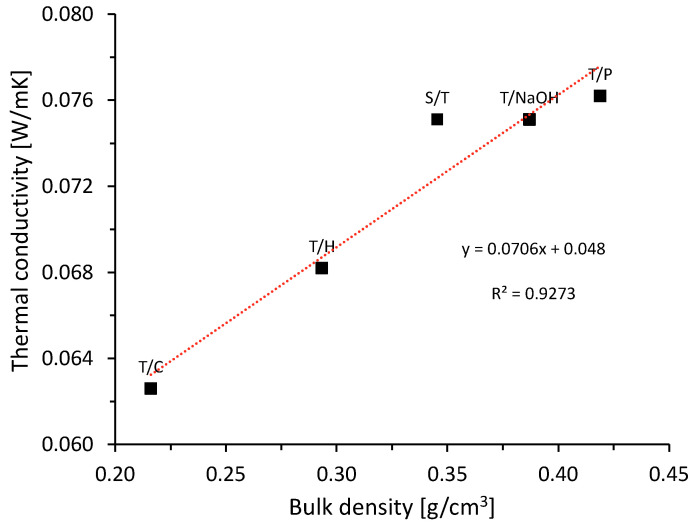
Relationship between bulk density and thermal conductivity of aggregates with different treatments.

**Figure 10 materials-15-06000-f010:**
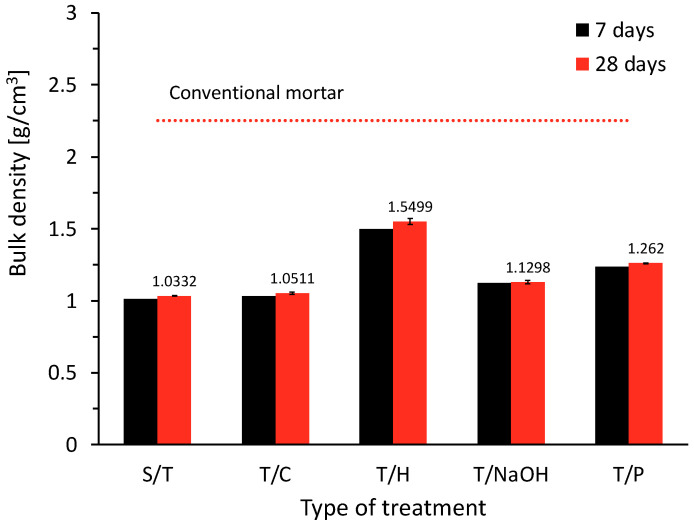
Bulk density on PBC specimens.

**Figure 11 materials-15-06000-f011:**
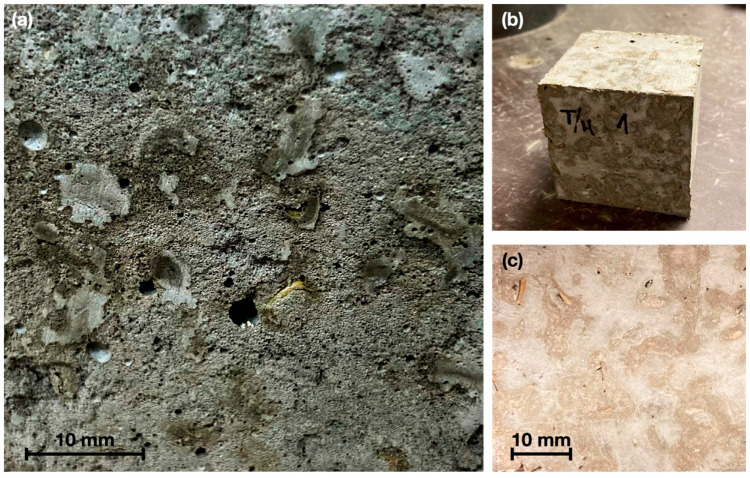
(**a**) plant-based concrete sample with an abundant presence of air bubbles due to saponins in the vegetal aggregate; (**b**,**c**) densified cementitious matrix for T/H.

**Figure 12 materials-15-06000-f012:**
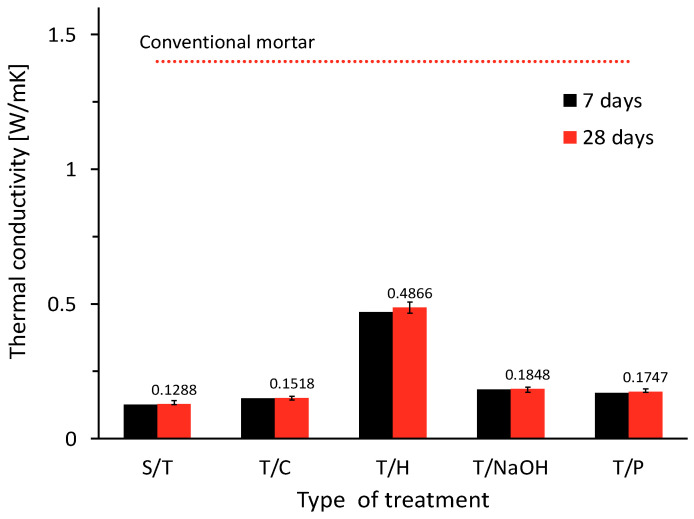
Thermal conductivity results from vegetal concrete with aggregate subjected to different treatments.

**Figure 13 materials-15-06000-f013:**
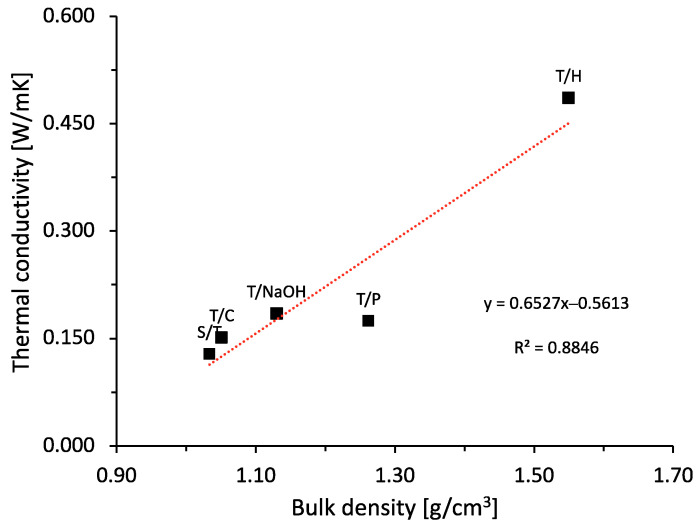
Relation between bulk density and thermal conductivity.

**Figure 14 materials-15-06000-f014:**
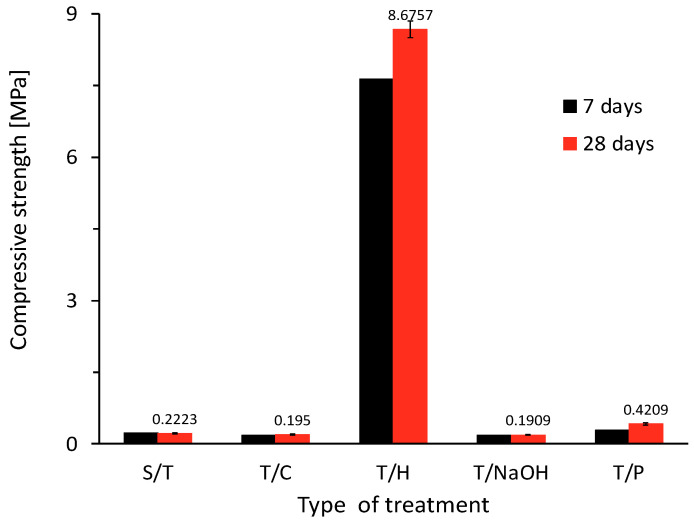
Compressive strength test results for plant-based concrete with treatments.

**Figure 15 materials-15-06000-f015:**
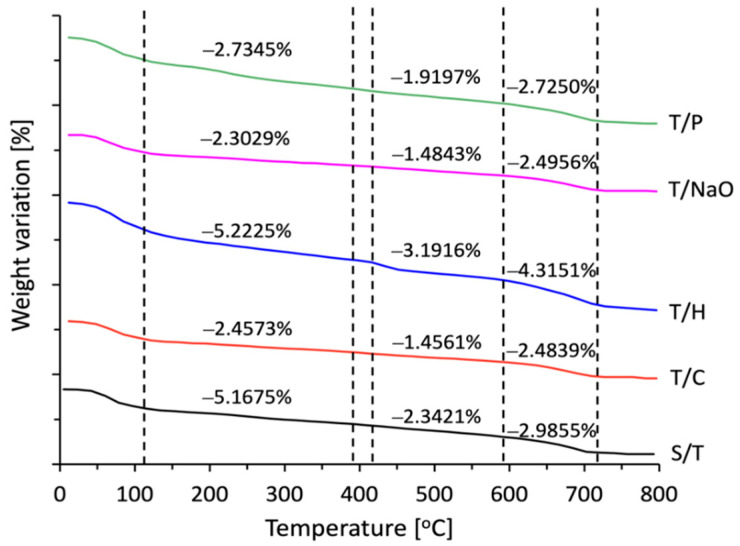
Comparative graph of TGA of cement paste for each of the cases under study of CV.

**Figure 16 materials-15-06000-f016:**
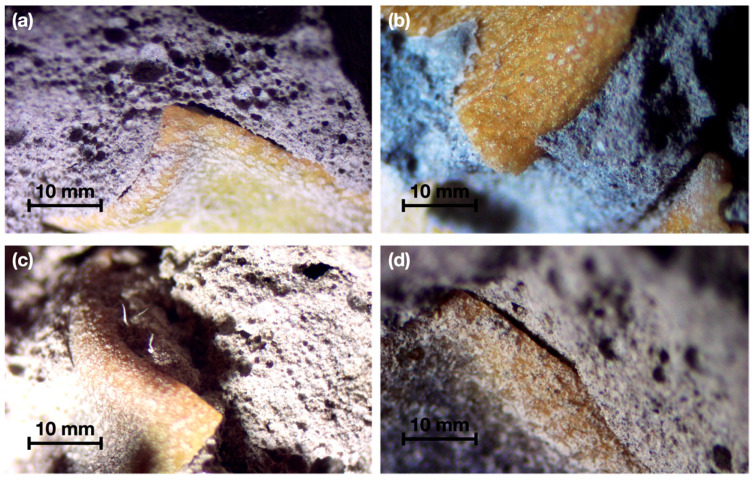
Microscopy performed on plant-based concrete samples. (**a**) S/T; (**b**) T/H; (**c**) T/NaOH; (**d**) T/P.

**Figure 17 materials-15-06000-f017:**
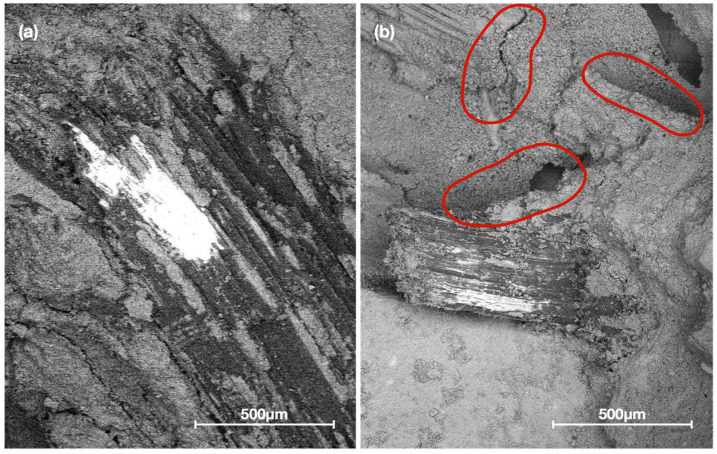
SEM micrographs of plant-based concrete. (**a**) Plant-based concrete with hornification treatment. (**b**) Plant-based concrete with untreated aggregate showing the detachment between bioaggregate and matrix.

**Table 1 materials-15-06000-t001:** Advantages and disadvantages of the use of bioaggregates in plant-based concretes.

Advantages	Disadvantages
Low specific weight	Low mechanical resistance
Renewable resource	Variable fiber quality
Production with low investment and costs	Low moisture resistance
High electrical resistance	Low durability
Good thermal and acoustic insulation	Low fire resistance
Biodegradable	Low adhesion between fiber and matrix

**Table 2 materials-15-06000-t002:** XRF test results of CPO40R.

Composite	MgO	Al_2_O_3_	SiO_2_	SO_3_	K_2_O	CaO	TiO_2_	Fe_2_O_3_
%	0.18	1.43	20.71	6.09	1.36	66.78	0.29	2.78

**Table 3 materials-15-06000-t003:** Dosage of plant-based concrete mixes (AV: Plant aggregate; AP: Premix water; OPC: Cementitious; AR: Reaction water).

Sample	AV [g]	AP [g]	OPC [g]	AR [g]
S/T	46.6155	21.8120	546	191.1
T/C	29.1735	19.7192	546	191.1
T/H	39.6090	23.8059	546	191.1
T/NaOH	52.2315	23.0637	546	191.1
T/P	56.5515	20.1366	546	191.1

**Table 4 materials-15-06000-t004:** Results of chemical characterization of *Agave salmiana*.

Test	Average
Moisture analysis	5.20 ± 0.03%
Ash analysis	15.77 ± 1.90%
Ethanol-Toluene Extractables	23.47 ± 2.45%
Acid insoluble lignin	10.61 ± 0.24%
Holocellulose content	42.64 ± 1.10%
Cellulose content	49.64 ± 0.83%
Hemicellulose content	50.36 ± 0.83%

**Table 5 materials-15-06000-t005:** Results obtained for bulk density and thermal conductivity for each identified part of the AS leaf.

Samples	ρ [g/cm^3^]	λ [W/mK]
Pith	0.2994	0.0682
Pith with Cuticle	0.3350	0.0733
Cuticle	0.3771	0.0810

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
