# Peer review of "Development of a Portland Cement-Based Material with Agave salmiana Leaves Bioaggregate"

_materials, 2022, doi:10.3390/ma15176000_

Round 1

Reviewer 1 Report

In my opinion manuscript materials-1871790 is well written and deserves publication.

Some suggestion to improve the manuscript:

1.       Emphasize novelty at the end of the abstract.

2.       Mechanical properties are actually physical properties (lines 18-19)

3.       Correct x-axis legend in Figure 2.

4.       Pith is the same with marrow and cuticle with bark? Use consistent notions, see fig 5.

5.       Give scale bars in Fig 4 – left vertical images, Figure 9 and Figure 13.

6.       Bulk density is the same with volumetric weight? (fig 7, 8 and 11)

7.       Correct typesetting: 8.66MPa should be 8.66 MPa, [1][2] should be [1,2], etc.

Reviewer 2 Report

Dear Authors, 

Thank you for your manuscript. Here will be following remarks to it:

1. Figure 1 should be updated, to look more attractive&creative for a reader;

2. Introduction is insuficient, missing structure and related specific information in regard the research subject is rather limited. It is suggested to elaborate intro in a more efficient way and underline the novelty of the study. 

3. Figure 3 it is nice image but it should also contain the plant aggregates image to see how it looks before it is applied in the mix. 

4.lines 116-123: here you can provide a Reader a sheme of the production, please elaborate a nice scheme for a better visualization of what you are doing to obtain aggregates;

5. Line 139: there should be some intro paragraph to summarize all methods for treatment;

6. Please make a figure where you indicate pros and cons of aggregates application in cementitious materials;

7. Table 5, provide instead a chart here please; reference mixture results? Treatments abbreviation has to be incorporated in chart;

Paper is missing clear structure: 1) statement of current problems / research questions related to deficiency of aggregates (which regions, where plant aggregates can be applied, the volumes of plant aggregates, production cost in comparison to the natural ot other alternative aggregates, 2) if reference 38 is taken as a base for this research what is novelty of your study? There should be clear identification of the pros&cons of ref 38 & the pros of current paper; 3) pros & cons of plant aggregates applications; 4) detailed /  of the art of the previous research and what is novel in current paper; 5) conclusions must be elaborated more in detail based on the results; 6) SEM images would be a must for this paper to see the ITZ in the matrix of cementitious material. 

The paper more looks like now as a good start draft, please elaborate it more with a scientific/valorization input. 

Round 2

Reviewer 2 Report

Dear Authors, thank you for considering my remarks, now your paper looks very good and hopefully you will get a good citations record for it.